# Beyond Overconfidence: Rethinking Calibration in Large-Scale Vision Models

## Abstract

Reliable uncertainty calibration is crucial for the safe deployment of deep neural networks in high-stakes settings. While these networks are known to exhibit systematic overconfidence, especially under distribution shifts, the calibration of large-scale vision models, such as ConvNeXt, EVA, and BEiT, has remained underexplored. We comprehensively examine their calibration behavior, uncovering evidence that challenges well-established assumptions. We find that these models are underconfident on in-distribution data, which results in increased calibration error, yet exhibit improved calibration under distribution shifts. This phenomenon is primarily driven by modern training techniques, including massive pretraining and sophisticated regularization and augmentation methods, rather than architectural innovations alone. We also demonstrate that these large-scale models are highly responsive to post-hoc calibration techniques in the in-distribution setting, enabling practitioners to mitigate underconfidence bias effectively. However, these methods become progressively less reliable under severe distribution shifts and can occasionally produce counterproductive effects. Our findings highlight the complex, non-monotonic effects of architectural and training innovations on calibration, challenging established narratives of continuous improvement.

## 1    Introduction

Deep neural networks deployed in high-stakes applications require not only high predictive accuracy but also reliable uncertainty estimates. In safety-critical domains, such as medical diagnosis, autonomous driving, and financial decision-making, the consequences of incorrect predictions accompanied by misleadingly high confidence scores can be severe. Model calibration – aligning predicted confidence with empirical accuracy – provides a formal framework for assessing the reliability of these uncertainty estimates (Guo et al., 2017). In a well-calibrated model, predictions made with 80% confidence should be correct approximately 80% of the time.

A fundamental challenge in model calibration is that deep neural networks are typically poorly calibrated. In particular, they tend to exhibit systematic overconfidence, assigning probabilities to predictions that exceed their actual accuracy (see, e.g., Guo et al. (2017); Hendrycks et al. (2021); Lakshminarayanan et al. (2017); Rahaman & Thiery (2021); Cheng & Vasconcelos (2024); Wang et al. (2021)). This calibration error becomes even more pronounced when models encounter distribution shifts (see, e.g., (Ovadia et al., 2019; Hendrycks & Dietterich, 2019)). To address the challenge of miscalibration, post-hoc calibration methods (see, e.g., Guo et al. (2017); Zhang et al. (2020b); Gupta et al. (2021); Tomani et al. (2022)) are promising since they can be applied directly to trained models and can therefore be used as a lightweight post-processing step to recalibrate the model's outputs. Additionally, Minderer et al. (2021) demonstrated that architectural innovations available at the time (such as Vision Transformers) have inherently well-calibrated outputs and improved robustness to distribution shifts, suggesting that miscalibration is more pronounced in traditional models than in then-current state-of-the-art models.

Most recent advances in deep learning have catalyzed the emergence of models featuring large-scale training regimes, characterized by massive-scale pre-training using novel training techniques (e.g., masked image modeling (He et al., 2022; Bao et al., 2021)) and sophisticated regularization and augmentation techniques (e.g., CutMix (Yun et al., 2019), MixUp (Zhang et al., 2017), label smoothing (Zhang et al., 2020a; Lukasik et al., 2020), and RandAugment (Cubuk et al., 2020)).

While models trained with these approaches – such as ConvNeXt, EVA, and BEiT – achieve state-of-the-art accuracy and have led to a broad adoption by practitioners, the implications for model calibration properties remain insufficiently explored. Specifically, it is unclear whether exposure to diverse, web-scale training data improves calibration by providing broader coverage of the input distribution, or if it introduces new calibration issues due to inherent dataset biases and complex regularization schemes. Furthermore, despite significant advances in post-hoc calibration techniques for traditional neural architectures, the efficacy of these techniques when applied to these large-scale models has not been adequately investigated.

In this paper, we systematically benchmark the quality of predictive uncertainty of large-scale vision models and make the following key contributions:

1. Through a systematic benchmark, we demonstrate that large-scale models (ConvNeXt, EVA, and BEiT) exhibit significant in-distribution calibration errors, characterized by systematic **underconfidence** in predictive probabilities. This finding contrasts with the well-documented overconfidence bias observed in traditional deep neural networks.

2. Our analysis reveals that this systematic underconfidence in large-scale models results from the combination of pretraining on extensive datasets and advanced regularization strategies, rather than architectural design choices.

3. We further show that large-scale models maintain calibration quality under both synthetic and real-world distribution shifts. This finding contrasts with traditional neural architectures, which exhibit a monotonic increase in calibration error as the magnitude of the distribution shift increases.

4. We demonstrate that post-hoc calibration methods can significantly improve the calibration of large-scale models for in-distribution predictions. However, their benefits diminish under distribution shift.

## 2 RELATED WORK

**Empirical Studies of Model Calibration**    Over the past decade, research into neural network calibration has established a strong empirical foundation. The seminal work by Guo et al. (2017) first documented that neural architectures used at the time, such as ResNets and DenseNets, typically produce overconfident predictions. Several subsequent studies have corroborated this finding (Thulasidasan et al., 2019; Hendrycks et al., 2021; Lakshminarayanan et al., 2017; Rahaman & Thiery, 2021).

Distribution shift conditions exacerbate these calibration issues: Ovadia et al. (2019) demonstrated through a comprehensive evaluation that "along with accuracy, the quality of uncertainty consistently degrades with increasing dataset shift." Hendrycks & Dietterich (2019) further validated this phenomenon, whose ImageNet-C benchmark revealed a direct correlation between corruption severity and increasing calibration error. Similarly, Recht et al. (2019) demonstrated that temporal distribution drift in ImageNet-V2 negatively impacts both predictive performance and calibration metrics.

Recent architectural advancements have challenged these established patterns. Minderer et al. (2021) documented improved calibration in Vision Transformers and MLP-Mixers compared to previous generations of models. They noted that these models were "well calibrated compared to past models and their performance is more robust to distribution shift." They also emphasized the importance of model architecture in determining calibration quality, suggesting that the most recent architectural innovations may improve calibration quality inherently. Tao et al. (2024) further substantiated this architectural dependency. Their large-scale calibration benchmark, which used NAS-searched architectures, demonstrated a strong correlation between the design choices of neural networks and their calibration properties. However, their investigation was limited to models with conventional training regimes and did not examine large-scale models pre-trained on massive datasets.

Recent work has emphasized decomposing predictive uncertainty to understand miscalibration sources beyond aggregate metrics. Perez-Lebel et al. Perez-Lebel et al. (2023) show that even perfectly calibrated classifiers can exhibit grouping loss—samples with identical confidence but

different true probabilities. This builds on proper scoring rule theory Murphy (1973a); Bröcker (2009); Kull & Flach (2015), which decomposes predictive errors into calibration, refinement, and irreducible components.

**Post-hoc Calibration Techniques** Post-hoc calibration methods represent a computationally efficient framework for enhancing the reliability of neural network confidence estimates without requiring architectural modifications or extensive retraining procedures. These approaches operate by learning mapping functions that transform a model's raw outputs into recalibrated probability distributions, thereby optimizing the correspondence between predictive confidence and empirical accuracy. The re-calibration process typically leverages a held-out validation set to estimate the parameters of these transformations while maintaining the model's discriminative capabilities.

The literature has explored various approaches to post-hoc calibration, each of which is characterized by a distinct set of trade-offs between functional expressivity, parameter efficiency, and generalization properties Guo et al. (2017); Zhang et al. (2020b); Gupta et al. (2021); Kull et al. (2019). While post-hoc calibration methods achieve strong performance on in-distribution data, their reliability degrades substantially under distribution shift. Recent advances address this limitation through density-aware approaches that ensure calibration not only globally but also within local regions of the input space Xiong et al. (2023); Tomani et al. (2023).

## 3 PROBLEM DEFINITION AND NOTATION

In this paper, we systematically benchmark the quality of predictive uncertainty of multi-class neural network models across different architecture and training paradigms. A neural network parameterizes a prediction function $f$ that maps input $\boldsymbol{x} \in \mathbb{R}^D$ to a probability vector $\boldsymbol{p} \in [0,1]^C$ over $C$ classes. These predictions reside in the $(C-1)$-dimensional probability simplex: $\triangle = \{\boldsymbol{p} \in [0,1]^C \mid \sum_{c=1}^{C} p_c = 1\}$, where $p_c$ denotes the $c$-th component of the probability vector $\boldsymbol{p}$.

A model $f$ is perfectly calibrated (Bröcker, 2009) if and only if:

$$\forall \boldsymbol{p} \in \triangle : \mathbb{P}(y = c \mid f(\boldsymbol{x}) = \boldsymbol{p}) = p_c . \tag{1}$$

Throughout this paper, we focus on the weaker notion of *top-label calibration* (Guo et al., 2017), requiring that predictions made with maximum confidence $p^* = \max f(\boldsymbol{x})$ are correct with probability $p^*$:

$$\forall p^* \in [0,1] : \mathbb{P}(y \in \arg\max f(\boldsymbol{x}) \mid \max f(\boldsymbol{x}) = p^*) = p^* . \tag{2}$$

To quantify top-label calibration error, we compute the Expected Calibration Error (ECE), which measures the expected discrepancy between the two sides of Eq. 2 and is defined as

$$\mathbb{E}[|p^* - \mathbb{P}(y \in \arg\max f(\boldsymbol{x}) \mid \max f(\boldsymbol{x}) = p^*)|] . \tag{3}$$

Due to the continuous-valued probability space, direct estimation of Eq. 3 is intractable. Therefore, a binning approach is typically employed by partitioning the prediction space into $m$ equally spaced bins $B_1, \ldots, B_m$. Given $n$ i.i.d. samples $(\boldsymbol{x_i}, y_i)_{i=1}^{n}$ drawn from the joint distribution $\mathbb{P}(\boldsymbol{x}, y)$, we assign each $i \in \{1, \ldots, n\}$ to a bin $B_j$ based on $\max f(\boldsymbol{x_i})$.

Then, we compute for each bin $B_j$ the mean top-level confidence $\text{conf}(B_j) = \frac{1}{|B_j|} \sum_{i \in B_j} \max f(\boldsymbol{x_i})$ and the mean accuracy $\text{acc}(B_j) = \frac{1}{|B_j|} \sum_{i \in B_j} \mathbf{1}(\arg\max f(\boldsymbol{x_i}) = y_i)$ and finally compute the Expected Calibration Error according to

$$\text{ECE} = \sum_{j=1}^{m} \frac{|B_j|}{n} |\text{acc}(B_j) - \text{conf}(B_j)| . \tag{4}$$

In addition to ECE, we quantify Brier score and the negative log likelihood as proper scoring rules, capturing both model calibration and model sharpness Murphy (1973b); Popordanoska et al. (2024). Formal definitions of these metrics are provided in the supplementary material.

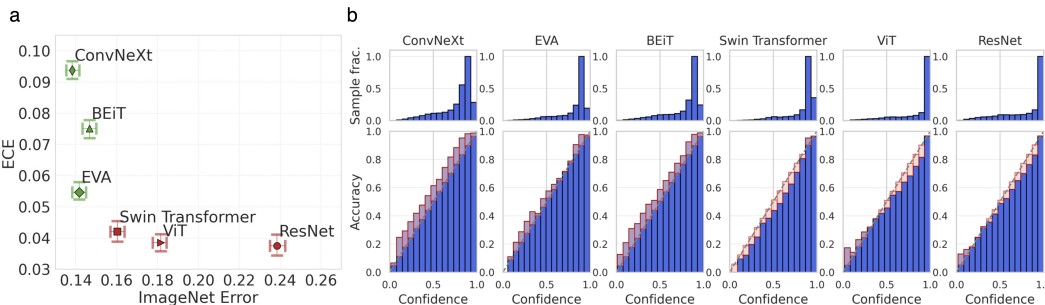

Figure 1: (a) Inverse relationship between ImageNet classification error and Expected Calibration Error (ECE). Green markers represent large-scale models (from 2022), while red markers represent traditional models (until 2021). Error bars show 95% bootstrap confidence intervals (n=100). Despite their superior classification performance, large-scale models consistently exhibit significantly higher calibration errors. (b) Reliability diagrams showing the systematic underestimation of predictive confidence in large-scale models (ConvNeXt, EVA, and BEiT), contrasting with the overconfidence observed in traditional models (Swin Transformer, ViT, and ResNet).

## 4 EMPIRICAL EVALUATION

### 4.1 EXPERIMENTAL SETUP

**Models Under Evaluation.** Throughout the paper we consistently evaluate six neural networks. We distinguish between traditional training paradigms (ResNet-50, ViT-B/16, Swin-S3-B) and contemporary large-scale training regimes characterized by massive pretraining combined with sophisticated regularization and augmentation techniques (BEiT-B/16, EVA-S/14, ConvNeXt-B). Note that this distinction is based on training methodology rather than model architecture or pretraining dataset scale alone. To disentangle the effects of architecture from training methodology, we additionally evaluate ViT and ResNet variants trained with large-scale techniques while maintaining traditional architectures. Detailed model specifications are provided in Appendix B.

**Datasets.** We evaluate accuracy and calibration error on the **ImageNet-1k** dataset Deng et al. (2009) and the following distributed-shifted benchmarks:

1. **ImageNet-C** (Hendrycks & Dietterich (2019)), which augments the standard ImageNet-1k dataset by introducing 19 distinct types of synthetic corruptions, each applied at 5 severity levels.
2. **ImageNet-V2** (Recht et al. (2019)), comprising 10,000 temporally shifted real-word samples collected using the original ImageNet-1k sampling protocol.
3. **ImageNet-A** (Hendrycks et al. (2021)), containing 7,500 natural adversarial examples specifically selected for their ability to induce misclassification in standard ResNet-50 models.

To optimize post-hoc calibration methods, 10% of the ImageNet-1k validation set is randomly selected for parameter tuning. All metrics are reported on the remaining 90% of the validation set to ensure methodological consistency. For ImageNet-C, we ensure that the images used for tuning the post-hoc calibration parameters are excluded from their corresponding corrupted versions, to prevent data leakage.

**Post-hoc Calibration Techniques.** To systematically evaluate the effectiveness of post-hoc calibration techniques across various models, we evaluate for post-hoc calibration methods: Temperature Scaling (TS), Ensemble Temperature Scaling (ETS), Isotonic Regression (IRM), and Spline Calibration (SPL). Details are provided in Appendix C.

**Calibration Metrics.** Throughout our analysis, we follow Minderer et al. (2021) and estimate the Expected Calibration Error (ECE) using 15 equal-mass bins as our primary calibration metric. To

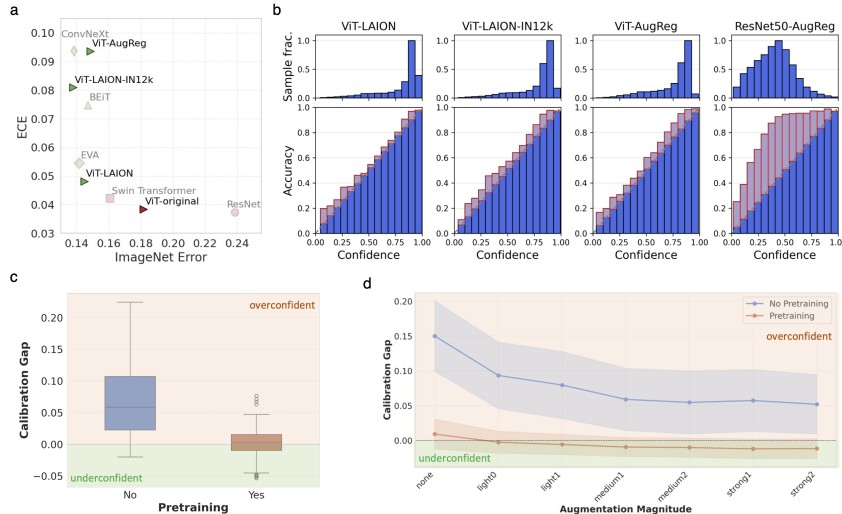

Figure 2: (a) ECE vs. classification error shows that the same ViT architecture trained with modern training techniques (green markers) have higher calibration errors than the original ViT model (red marker). Grey markers represent models from previous experiments that are included for comparison. (b) Reliability diagrams reveal systematic underconfidence across models pretrained on large datasets (ViT-LAION and ViT-LAION-IN12k) or aggressive regularization and augmentation techniques (ViT-AugReg and ResNet50-AugReg). (c) Box plot of calibration gap (difference between mean predicted confidence and accuracy) across 1008 ViT models confirms that pretraining systematically shifts models toward underconfidence. (d) Calibration gap as a function of augmentation magnitude demonstrates that increasing augmentation magnitude results in more underconfident models.

provide a more comprehensive analysis, we present additional results in the supplementary material, using different bin sizes and alternative metrics (e.g. Brier score and negative log-likelihood as proper scoring rules).

## 4.2 LARGE-SCALE MODELS EXHIBIT SYSTEMATIC IN-DISTRIBUTION UNDERCONFIDENCE

First, we investigate the inherent in-distribution calibration properties of neural networks, before applying any post-hoc calibration techniques. Unlike the findings of Minderer et al. (2021), who reported concurrent improvements in accuracy and calibration for then-current models, our investigation reveals a significant divergence in this relationship for contemporary large-scale models (Figure 1a). While recent model innovations have substantially improved classification performance, they have also demonstrated an increasing calibration error, showing an emerging trade-off between these performance aspects.

Closer examination of the reliability diagrams (Figure 1b) reveals that the increased ECE of the large-scale models is due to a systematic underconfidence in in-distribution predictions – a notable departure from the widespread overconfidence documented in previous calibration literature.

Although this underconfidence increases the overall calibration error, it indicates a different calibration regime that could be advantageous for deployment in high-stakes domains.

## 4.3 EXPLORING FACTORS INFLUENCING CALIBRATION BEHAVIOR

While our previous experiments reveal systematic underconfidence in large-scale models, the underlying mechanisms driving these phenomena remain unclear. To gain insights into these mechanisms, we conduct a controlled experiment that isolated the influence of training methodology from architectural design.

In this experiment, we utilize the same Vision Transformer architecture as in the previous experiments, while systematically varying the training settings. Specifically, we first explore different

pre-training pathways by pre-training the Vision Transformer on the large-scale LAION dataset (Cherti et al., 2023). Then the model is further trained either by (1) direct fine-tuning on ImageNet-1k (ViT-LAION) or (2) sequential fine-tuning on ImageNet-12k and then fine-tuned on ImageNet-1k (ViT-LAION-IN12k). Second, we investigate the impact of aggressive augmentation and regularization techniques for the ViT (ViT-AugReg) and the ResNet50 architecture (ResNet50-AugReg) while maintaining the same dataset for training as the original models (Steiner et al., 2022).

As illustrated in Fig. 2a, both methodological variations substantially improve the classification accuracy for the ViT models while concurrently increasing ECE.

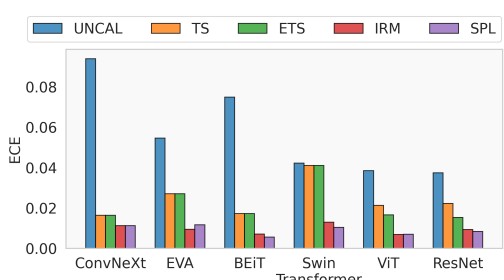

This accuracy-calibration trade-off is even more pronounced for ResNet50-AugReg, where modern training techniques boost accuracy from 76.2% to 80.4% but cause ECE to increase from 0.037 to 0.408, a tenfold increase in miscalibration that exemplifies how contemporary training practices can severely compromise model calibration. The reliability diagrams in Figure 2b reveal that these modern training methods induce systematic underconfidence in both architectures, transforming even traditional models like ResNet50 from their typical overconfident behavior to the underconfident regime characteristic of contemporary large-scale models.

Figure 3: Comparative evaluation of post-hoc calibration methods shows that simple temperature scaling (TS) is sufficient to align the calibration performance of large-scale models with that of traditional models.

To further strengthen our analysis on the causes underlying the observed underconfidence, we conducted a large-scale ablation study with 1,008 ViT models of different sizes as well as different pretraining, regularization, augmentation and fine-tuning strategies. Detailed parameter configurations and the complete ablation design are provided in Appendix B. This ablation study further supported our conclusions and revealed a consistent mechanism behind underconfidence: stronger augmentations induce underconfidence (see Fig. 2d), and pretraining consistently amplifies this effect across all ViT variants (see Fig. 2c).

Beyond the empirical results, Vicinal Risk Minimization (VRM) provides a theoretical framework to explain the underconfidence caused by stronger augmentations and pretraining. VRM minimizes the expected loss not only on the empirical samples $x$ but also on a neighborhood $v(x)$ induced by data augmentation:

$$\mathcal{L}_{\text{VRM}} = \mathbb{E}_{(x,y)\sim\mathcal{D}}\mathbb{E}_{x'\sim v(x)}\ell(f(x'), y) \tag{5}$$

Therefore, the model also assigns probability mass to neighborhoods around each training sample. When augmentations become stronger or more diverse (e.g., RandAugment, Mixup), the vicinal distribution becomes increasingly smoother and more dispersed, effectively enlarging the support region around each labeled example. This has two key implications relevant to the observed underconfidence: First, a wider vicinal distribution forces the classifier to assign similar probabilities to a broader set of augmented variants, which encourages smoother logits. This naturally pushes predictions toward underconfidence rather than overconfidence. Intuitively, the model learns to hedge its predictions across the enlarged neighborhood $v(x)$, resulting in lower confidence values even for correctly classified samples. Second, pretraining amplifies VRM's smoothing effect. Pretrained models already encode broader invariances and feature smoothness learned across large datasets. When combined with strong augmentations during fine-tuning, the model effectively samples from an even more dispersed vicinal distribution. This compounds the smoothing effect on logits, leading to stronger underconfidence - exactly what we observe empirically in our ablation study. Models pretrained on LAION-400M and subsequently fine-tuned with strong augmentation exhibit the most pronounced underconfidence, consistent with this compounding effect.

This theoretical viewpoint helps explain the universal trends across all 1,008 ViT models we evaluated and supports the conclusion that underconfidence emerges primarily from properties of the training strategy rather than from architectural biases. The systematic nature of our findings - spanning different model scales, augmentation strategies, and pretraining regimes - demonstrates that the

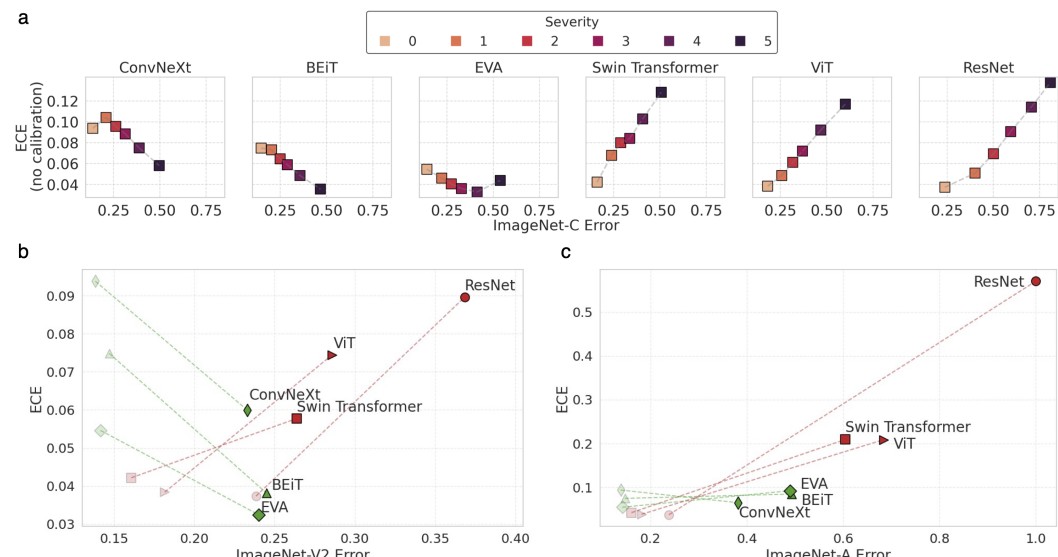

Figure 4: (a) Classification error and Expected Calibration Error (ECE) on ImageNet-C for the uncalibrated models. Severity 0 refers to the clean ImageNet test set. The calibration of large-scale models is more robust to distribution shift than past models. (b) Classification error and ECE on ImageNet-V2 and (c) on ImageNet-A, comparing large-scale models (green markers) with traditional models (red markers). For comparison, the performance of the same models on the clean (in-distribution) ImageNet test set is shown by grey markers of the same type, allowing the distribution shift effect to be visualised. Dotted lines connect each model's in-distribution result with its corresponding out-of-distribution result.

VRM perspective provides a unifying framework for understanding calibration behavior in modern vision models.

## 4.4 POST-HOC CALIBRATION FOR IN-DISTRIBUTION PREDICTIONS

Next, we evaluate the effectiveness of post-hoc calibration techniques in addressing the in-distribution miscalibration (Fig. 3).

Remarkably, the simple TS approach is sufficient to align the calibration performance of large-scale models with that of traditional models. However, while ETS theoretically offers greater flexibility by incorporating ensemble-based transformation, it provides no measurable benefits for large-scale models compared to simple TS, despite its higher expressive power.

Among the evaluated methods, isotonic regression (IRM) and spline calibration (SPL) achieve the best calibration quality, outperforming temperature-based approaches consistently across all architectures. This is likely due to their ability to learn a more flexible, nonlinear transformation of the confidence scores.

## 4.5 CALIBRATION UNDER DISTRIBUTION SHIFT

Building on our observation that large-scale models exhibit systematic underconfidence on in-distribution data, we now investigate how their calibration properties change when they are faced with synthetic and real-world distribution shifts.

**Synthetic Distribution Shifts** We first analyze model calibration under controlled synthetic corruptions using ImageNet-C (Figure 4a). While traditional models follow the well-documented pattern of decreasing accuracy and increasing calibration error as corruption severity increases, large-scale models exhibit a fundamentally different behavior. As the severity of corruption increases, these models show the expected drop in classification accuracy and a decrease in ECE values.

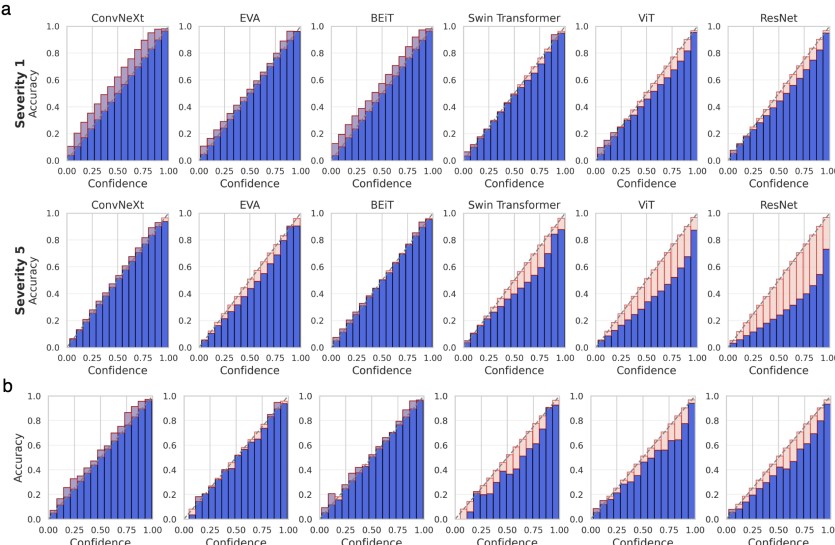

Figure 5: Reliability diagrams comparing predicted confidence with empirical accuracy for (a) artificial distribution shifts of ImageNet-C at severity levels 1 and 5, averaged across all 19 corruption types, and (b) real-world distribution shift using ImageNet-V2. Reliability diagrams for intermediate severity levels and ImageNet-A are provided in the supplementary material.

This counterintuitive improvement in calibration metrics can be explained by examining the underlying dynamics of confidence. Previous studies have shown that distribution shifts typically lead to an increase in model confidence compared to accuracy. For large-scale models that start with underconfidence, this shift-induced increase in confidence acts as a corrective mechanism, bringing predictions closer to actual accuracy levels.

The reliability diagrams in Figure 5a confirm these patterns: traditional models transition to extreme overconfidence, while large-scale models improve calibration while maintaining mild underconfidence.

**Real-World Distribution Shift**  To validate whether these findings generalize beyond synthetic perturbations, we examine calibration behavior under real-world distribution shifts using ImageNet-V2 and ImageNet-A (Figure 4b and 4c).

On ImageNet-V2, large-scale models demonstrate consistent improvements in calibration, with ECE decreasing between 36% and 49%. In contrast, traditional models suffer from an increase in ECE ranging from 40% to 140%, reflecting their strong overconfidence under distribution shift (Figure 5b).

The severe distribution shift in ImageNet-A further amplifies these differences. While traditional models experience substantial calibration degradation, large-scale models maintain relatively stable calibration. Notably, ConvNeXt achieves a slight ECE improvement despite the highly challenging nature of the shift.

## 4.6 POST-HOC CALIBRATION UNDER DISTRIBUTION SHIFT

Finally, we evaluate the performance of post-hoc calibration methods under varying levels of distribution shift. As shown in Figure 6, temperature scaling consistently reduces the ECE for ViT and ResNet50 compared to the uncalibrated baseline as expected. However, the recalibration behavior for large-scale models is different, with its effectiveness dependent on the severity of the distribution shift. While these methods can significantly improve calibration under in-distribution conditions and mild corruptions (severity levels 1-2), their effectiveness decreases as the severity of the distribution shift increases. We observe that the performance of post-hoc calibration methods can degrade under severe distribution shifts to levels worse than those of uncalibrated models. However, overall the

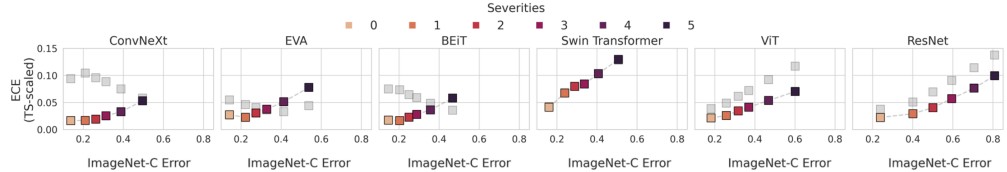

Figure 6: Analysis of classification error and Expected Calibration Error (ECE) across five severity levels of synthetic distribution shifts in ImageNet-C. While temperature scaling perform well under in-distribution conditions, its effectiveness declines with increasing shift magnitude, becoming even counterproductive. The gray markers represent uncalibrated results for comparison.

calibration error of large-scale models under distribution shift remains comparable or smaller than the CE of traditional models.

One underlying cause of this phenomenon is that large-scale models are calibrated on underconfident predictions from the in-distribution validation set, resulting in an increased global confidence. The subsequent application of TS to partially overconfident out-of-distribution samples then further exacerbates this overconfidence. However, observed behaviour of re-calibrated large scale models under distribution shift cannot be attributed solely to underconfidence. Our results reveal model-dependent responses to post-hoc calibration that follow patterns more complex than simple underconfidence correction would predict. EVA, for instance, exhibits distinct temperature scaling behavior at low shift severities that cannot be explained solely by its underconfidence pattern, demonstrating that post-hoc calibration effectiveness depends on model-specific confidence mechanisms beyond global underconfidence.

These findings are consistently observed across additional calibration metrics (ETS, IR, and SPL) as well as on real-world distribution shift datasets including ImageNet-V2 and ImageNet-A (see Appendix).

## 5 LIMITATIONS AND TAKEAWAYS

While our analysis provides robust evidence of systematic underconfidence in large-scale models, our focus in this paper is primarily on characterizing these phenomena rather than fully investigating their underlying causes. Although we conducted experiments in Section 4.3 to begin exploring these factors, these initial investigations could be extended in the future. Furthermore, our study focuses on vision-only classification models. Extending this analysis to vision-language models (e.g., LLaVA, Qwen-VL) represents an important direction for future work, though such models require different evaluation protocols due to their generative nature.

Based on our findings, we offer the following key insights for researchers and practitioners:

- **Underconfidence**: Large-scale models exhibit systematic underconfidence, which provides a practical advantage in safety-critical applications, as conservative uncertainty estimates reduce the risk of errors arising from overconfidence.

- **Distribution Shift Robustness**: Large-scale models demonstrate robust calibration under distribution shifts, ensuring that practitioners can deploy them in dynamic environments where data distributions evolve over time.

- **Limitations of Recalibration Techniques**: Post-hoc calibration methods can lead to counterproductive results for severe distribution shifts in large-scale models, so practitioners should exercise caution when applying these techniques in dynamic environments.

- **Determinants of Calibration Properties**: Calibration properties are predominantly determined by the specifics of the training procedures, rather than by architectural design choices.

- **Best Practices for Model Selection**: Among the evaluated models, ConvNeXt shows the most favorable trade-off between accuracy and calibration, providing comparatively reliable uncertainty estimates after recalibration, even under distribution shifts.

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

# APPENDIX

## A  SUMMARY

To support and expand upon our core findings, we provide additional metrics, experimental results, and technical details.

Section B provides comprehensive technical details of all models evaluated in this study, including the six primary benchmark models and the 1,008 ViT models trained for our ablation study. This section includes timm specifiers for reproducibility and model complexity metrics.

Section C describes the post-hoc calibration methods investigated in our study: Temperature Scaling (TS), Ensemble Temperature Scaling (ETS), Accuracy-Preserving Isotonic Regression (IRM), and Spline Calibration (SPL).

Section D introduces calibration metrics that complement the Expected Calibration Error (ECE) used in the main paper. These include Maximum Calibration Error (MCE), Root Mean Square Calibration Error (RMSCE), Root Brier Score (RBS), and Negative Log-Likelihood (NLL). These metrics capture different aspects of calibration quality and help validate the robustness of our findings.

Section E presents further in-distribution calibration results, demonstrating that the inverse relationship between classification and calibration errors holds across different ECE configurations, bin resolutions, and alternative calibration metrics. This section also includes additional results for post-hoc calibration techniques under distribution shift.

Section F presents reliability diagrams illustrating model calibration under varying degrees of distribution shift, including ImageNet-C at intermediate severity levels (2, 3, and 4), ImageNet-A, and individual reliability diagrams for all 19 synthetic corruptions at severity levels 3 and 5.

Section G evaluates the effectiveness of post-hoc calibration methods under real-world distribution shifts using the ImageNet-V2 and ImageNet-A datasets, confirming the patterns observed on ImageNet-C.

Section H provides a statement on LLM usage during manuscript preparation.

Sections I and J cover the availability of the code and the publicly available datasets that can be used to reproduce the experimental results and provide opportunities for further research.

# B    MODEL SPECIFICATIONS

This section provides technical details for all models evaluated in this study: our primary benchmark models and the 1,008 ViT models trained for our ablation study.

## B.1    BENCHMARK MODELS

Throughout the paper, we consistently analyze the following six models:

- **ResNet-50** (He et al. (2016)): A widely used baseline convolutional architecture, trained on ImageNet-1k.
- **ViT-B/16** (Dosovitskiy et al. (2021)): A pure transformer architecture pretrained on ImageNet-21k with supervised learning and fine-tuned on ImageNet-1k. Previous studies have demonstrated that this architecture has strong calibration properties.
- **Swin-S3-B** (Liu et al. (2021)): A hierarchical transformer model with shifted window partitioning, trained on ImageNet-1k.
- **BEiT-B/16** (Bao et al. (2021)): A transformer leveraging self-supervised masked image modeling, pretrained on ImageNet-22k and fine-tuned on ImageNet-1k.
- **EVA-S/14** (Fang et al. (2022)): A scaled transformer model pretrained on ImageNet-22k with self-supervised masked image modeling and subsequently fine-tuned on ImageNet-1k.
- **ConvNeXt-B** (Liu et al. (2022)): A convolutional architecture that integrates transformer-inspired design principles, pretrained on ImageNet-22k and fine-tuned on ImageNet-1k.

To investigate the factors contributing to underconfidence, we additionally evaluate several variants of ViT and ResNet models that employ traditional architectures but are trained with contemporary methodologies:

- **ViT-B/16-LAION**: A Vision Transformer pretrained on the LAION-2B dataset (2 billion image-text pairs) using contrastive learning objectives, subsequently fine-tuned on ImageNet-1k for classification (Cherti et al. (2023)).
- **ViT-B/16-LAION-IN12k**: A Vision Transformer pretrained on LAION-2B through contrastive learning, followed by sequential fine-tuning first on ImageNet-12k and then on ImageNet-1k (Cherti et al. (2023)).
- **ViT-B/16-AugReg**: A Vision Transformer initially pretrained on ImageNet-21k, then fine-tuned on ImageNet-1k using extensive data augmentation strategies and advanced regularization techniques including Mixup, Cutmix, and dropout scheduling (Steiner et al. (2022)).
- **ResNet50-AugReg**: The classical ResNet-50 architecture trained on ImageNet-1k, incorporating advanced regularization and augmentation techniques such as CutMix, MixUp, Label Smoothing, and Random Erasing.

This section provides technical details about the neural network models used in our calibration study (Table 1 and Table 2). All models can be accessed through the PyTorch Image Models (timm) library (see Table 1 and `https://github.com/huggingface/pytorch-image-models`), with corresponding checkpoints available on the Hugging Face model hub (`https://huggingface.co/`). This allows for direct reproducibility of our results. Input preprocessing followed the standard procedures specified in each model's documentation, including normalization with ImageNet statistics and appropriate resizing. For additional technical specifications beyond what is provided here, we refer to the model cards available on Hugging Face under the corresponding model identifiers and the documentation of the timm library.

## B.2    ABLATION STUDY MODELS

Our comprehensive ablation study comprises 1,008 independently trained Vision Transformer models based on the training protocol from Steiner et al. (2022). We evaluate six architectural variants (ViT-Ti/16, ViT-S/16, ViT-S/32, ViT-B/16, ViT-B/32, ViT-L/16) and systematically vary training parameters to isolate factors influencing calibration behavior (Table 3).

Table 1: Timm specifiers of the models used in the study.

| Model | timm Specifier |
|---|---|
| ResNet-50 | `resnet50.tv_in1k` |
| ViT-B/16 | `vit_base_patch16_224.orig_in21k_ft_in1k` |
| Swin-S3-B | `swin_s3_base_224.ms_in1k` |
| BEiT-B/16 | `beit_base_patch16_224.in22k_ft_in22k_in1k` |
| EVA-S/14 | `eva02_small_patch14_336.mim_in22k_ft_in1k` |
| ConvNeXt-B | `convnext_base.fb_in22k_ft_in1k` |
| ViT-LAION | `vit_base_patch16_clip_224.laion2b_ft_in1k` |
| ViT-LAION-IN12k | `vit_base_patch16_clip_224.laion2b_ft_in12k_in1k` |
| ViT-AugReg | `vit_base_patch16_224.augreg2_in21k_ft_in1k` |

Table 2: Complexities of the models used in the study.

| Model | Params (M) | GMACs |
|---|---|---|
| ResNet-50 | 25.6 | 4.1 |
| ViT-B/16 | 86.6 | 16.9 |
| Swin-S3-B | 71.1 | 13.7 |
| BEiT-B/16 | 86.5 | 17.6 |
| EVA-S/14 | 22.1 | 15.5 |
| ConvNeXt-B | 88.6 | 15.4 |
| ViT-LAION | 86.6 | 16.9 |
| ViT-LAION-IN12k | 86.6 | 16.9 |
| ViT-AugReg | 86.6 | 16.9 |

For data augmentation, we investigate seven configurations combining RandAugment and Mixup. Each configuration is described by a triple $(l, m, \alpha)$, where $l$ specifies the number of RandAugment operations, $m$ controls augmentation magnitude, and $\alpha$ is the Mixup interpolation parameter.

Table 3: Search space of training parameters for the ablation study.

| Hyperparameter | Values |
|---|---|
| Pretraining | {None, ImageNet-21k} |
| Weight decay | {0.03, 0.1} |
| Stochastic Depth & Dropout | {(0, 0), (0.1, 0.1)} |
| Learning Rate | {0.01, 0.03} |
| Augmentation (see Table 4) | 7 configurations |

Table 4: Augmentation configurations.

| Setup | $l$ | $m$ | $\alpha$ |
|---|---|---|---|
| none | 0 | 0 | 0 |
| light0 | 2 | 0 | 0 |
| light1 | 2 | 10 | 0.2 |
| medium0 | 2 | 15 | 0.2 |
| medium1 | 2 | 15 | 0.5 |
| strong0 | 2 | 20 | 0.5 |
| strong1 | 2 | 20 | 0.8 |

## C    POST-HOC CALIBRATION METHODS

We investigate the following commonly used post-hoc calibration techniques:

- **Temperature Scaling** (TS, Guo et al. (2017)) recalibrates network outputs using a single learned parameter that rescales the model's pre-softmax logits.

- **Ensemble Temperature Scaling** (ETS, Zhang et al. (2020b)) extends TS by constructing a weighted ensemble of temperature-scaled prediction, raw model outputs, and a uniform distribution over all classes.

- **Accuracy-Preserving Isotonic Regression** (IRM, Zhang et al. (2020b)) learns a strictly monotonic calibration function by pooling prediction-label pairs across all classes.

- **Spline Calibration** (SPL, Gupta et al. (2021)) learns continuous, piecewise polynomial functions to recalibrate model outputs.

# D ADDITIONAL CALIBRATION METRICS AND THEIR DEFINITIONS

To validate the robustness of our findings presented in the main paper, we extend our analysis of model calibration using different bin configurations and complementary calibration metrics. While our primary investigation focused on Expected Calibration Error (ECE) with 15 bins, we demonstrate here that our conclusions hold consistently across the following calibration metrics:

1. **Maximum Calibration Error (MCE)** quantifies the worst-case miscalibration scenario by measuring the maximum discrepancy between confidence and accuracy across all bins:

$$\text{MCE} = \max_j(\text{acc}(B_j) - \text{conf}(B_j)).$$

2. **Root Mean Square Error (RMSCE)** penalizes larger calibration errors more heavily than ECE by using squared differences:

$$\text{RMSCE} = \sqrt{\sum_j^m \frac{|B_j|}{n}(\text{acc}(B_j) - \text{conf}(B_j))^2}.$$

3. **Root Brier Score (RBS)** measures the accuracy of probabilistic predictions:

$$\text{BS} = \sqrt{\frac{1}{n}\sum_{i=1}^n \sum_{c=1}^C (p_{i,c} - y_{i,c})^2},$$

where $p_{i,c}$ represents the predicted probability for class $c$ of sample $i$, and $y_{i,c}$ is the corresponding one-hot encoded ground truth label.

4. **Negative Log-Likelihood (NLL)** evaluates the quality of probabilistic predictions by measuring the likelihood of the true lables under the model's predicted distributions:

$$\text{NLL} = -\frac{1}{n}\sum_{i=1}^n \log(p_{i,y_i}),$$

where $p_{i,y_i}$ is the predicted probability for the true class $y_i$ of sample $i$.

# E    FURTHER RESULTS ON IN-DISTRIBUTION CALIBRATION

## E.1    RESULTS FOR IN-DISTRIBUTION CALIBRATION FOR DIFFERENT ECE CONFIGURATIONS AND TYPES OF CALIBRATION ERROR

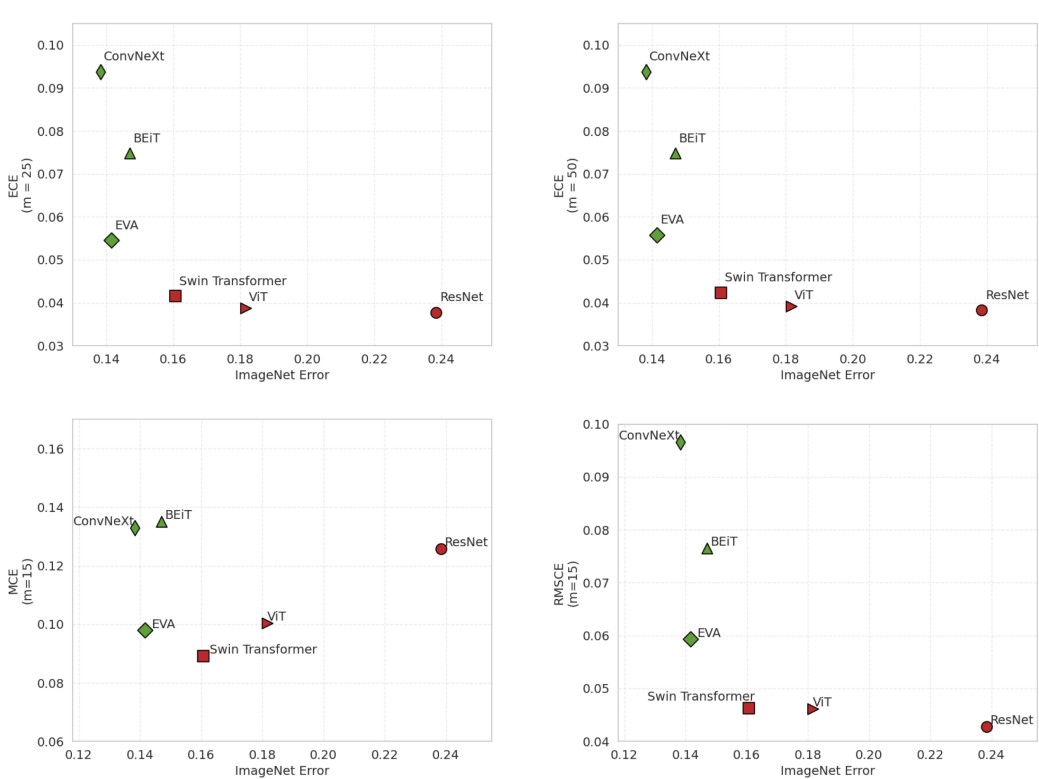

Figure 7: Scatter plot illustrating the inverse relationship between ImageNet classification error and calibration error. The results presented in the main part of the paper hold true for ECE across varying number of bins (m=25 and m=50) and for different types of calibration error, such as Maximum Calibration Error (MCE) and Root Mean Square Calibration Error (RMSCE).

## E.2 EFFECT OF BIN RESOLUTION ON RELIABILITY DIAGRAM

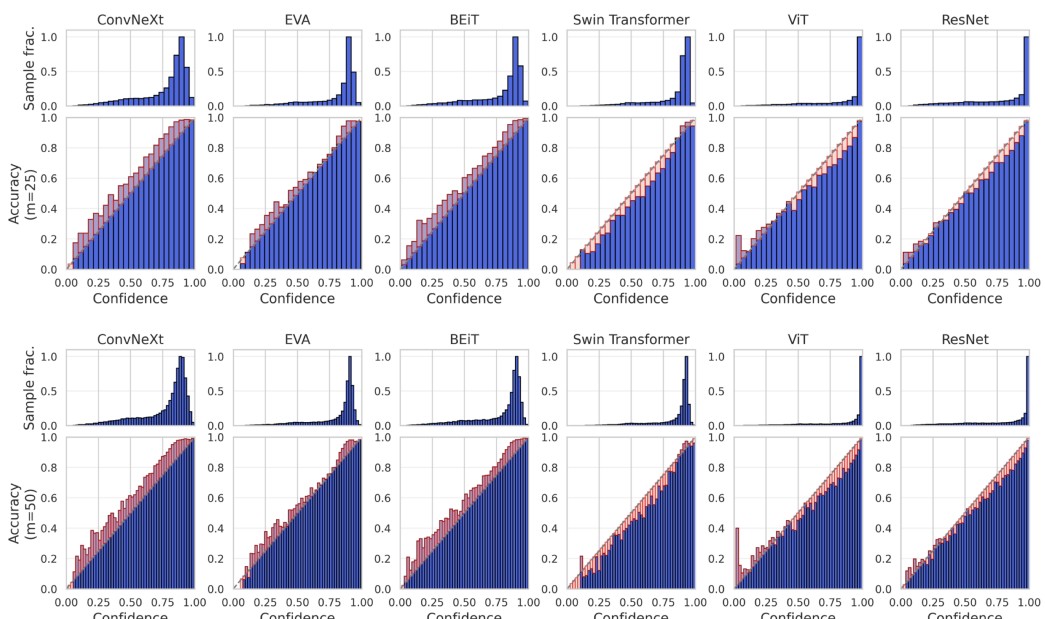

Figure 8: Reliability diagrams calculated with different bin resolutions (25 and 50 bins). The diagrams demonstrate that the observed calibration patterns remain consistent across different bin counts, supporting the robustness of our findings.

## E.3 RESULTS FOR POST-HOC CALIBRATION TECHNIQUES UNDER DISTRIBUTION SHIFT

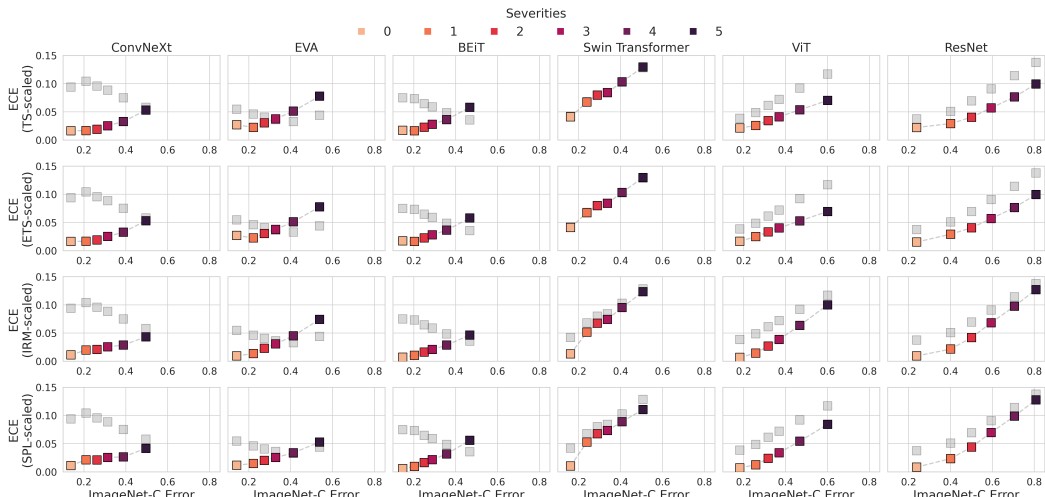

Figure 9: Additional results for the performance of ETS, IR and SPL under distribution shift.

### E.4 RESULTS FOR POST-HOC CALIBRATION TECHNIQUES FOR DIFFERENT CALIBRATION METRICS

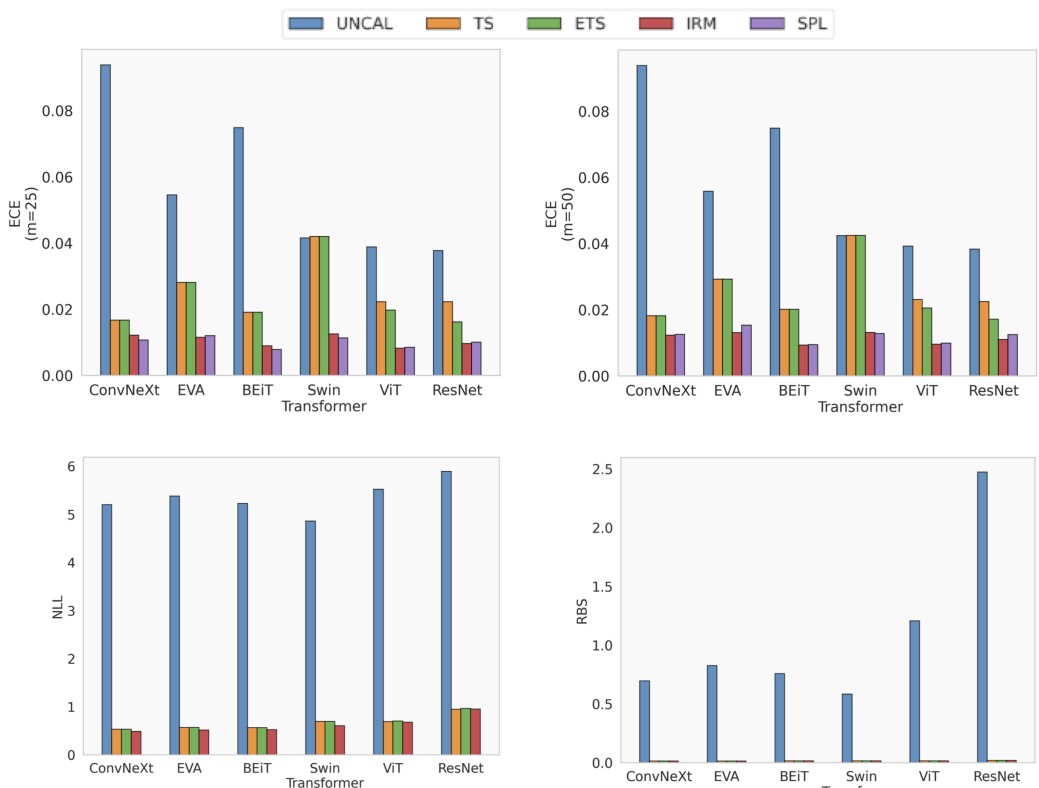

Figure 10: Comparison of post-hoc calibration effectiveness across multiple evaluation metrics: ECE with varying bin resolutions (25, and 50 bins), Root Brier Score (RBS), and Negative Log-Likelihood (NLL).

## F FURTHER RESULTS ON RELIABILITY DIAGRAMS UNDER DISTRIBUTION SHIFT

To provide a comprehensive view of model calibration behavior under varying distribution shifts, we additionally present the reliability diagrams for ImageNet-C at intermediate severity levels (2, 3, and 4) and for ImageNet-A (Figure 11)

These visualizations reveal the progressive changes in calibration behavior as distribution shift severity increases, demonstrating how large-scale models maintain their calibration advantage over traditional architectures. For traditional architectures (ResNet-50, ViT, and Swin), we observe a consistent pattern of increasing overconfidence as severity levels progress from 2 to 4. In contrast, foundation models (ConvNeXt, EVA, and BEiT) demonstrate remarkable robustness across these intermediate severity levels. Their initial underconfidence on in-distribution data gradually diminishes as severity increases.

The reliability diagrams for ImageNet-A complement our analysis of ImageNet-V2 presented in the main text and provide insights into calibration behavior under particularly challenging conditions. On ImageNet-A, traditional architectures exhibit extreme overconfidence across all confidence bins, with dramatic gaps between predicted probabilities and actual accuracy rates. Foundation models demonstrate significantly better calibrated predictions on ImageNet-A. In particular, ConvNeXt maintains relatively well-calibrated predictions across most confidence bins. These reliability diagrams further substantiate our findings that foundation models fundamentally alter the traditional calibration paradigm, maintaining better alignment between confidence and accuracy under challenging distribution shifts compared to traditional architectures.

Figures 12 and 13 provide a detailed view of calibration behavior across all 19 individual corruption types from ImageNet-C. Figure 12 presents reliability diagrams for each corruption type at severity level 3, while Figure 13 shows the corresponding diagrams at severity level 5.

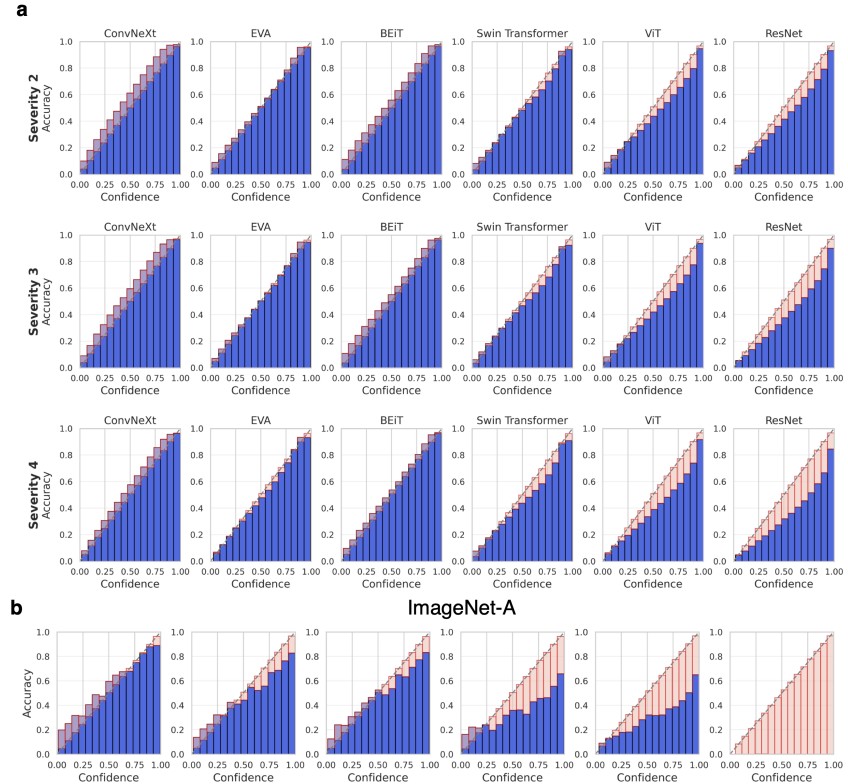

Figure 11: Reliability diagrams (m=15 bins) illustrating model calibration under (a) synthetic distribution shift induced by ImageNet-C corruptions at different severity levels (averaged over all corruptions), and (b) real-world distribution shift as represented by ImageNet-A.

## G  EVALUATING POST-HOC CALIBRATION PERFORMANCE UNDER REAL-WORLD DISTRIBUTION SHIFT

To validate our findings beyond synthetic corruptions, we extend our analysis to real-world distribution shifts using the ImageNet-V2 and ImageNet-A datasets (Figure 14). These benchmarks provide complementary perspectives on model robustness: ImageNet-V2 represents a moderate temporal distribution shift, while ImageNet-A introduces severe natural adversarial examples.

The ConvNeXt model demonstrates strong recalibration performance on ImageNet-V2, with post-hoc methods achieving significant ECE reductions. However, as the distribution shift becomes more severe on ImageNet-A, this effectiveness diminishes substantially, with recalibration methods yielding ECE values comparable to or exceeding those of the uncalibrated baseline. This pattern mirrors our observations with synthetic corruptions, where recalibration performance degraded with increasing severity.

EVA exhibits even more pronounced calibration challenges. Even under moderate shifts (ImageNet-V2), post-hoc calibration methods not only fail to improve calibration but actively increase ECE compared to the uncalibrated model. This aligns with trends observed under synthetic corruptions, where EVA's recalibration performance began deteriorating at lower severity levels than other foundation models. On ImageNet-A, all methods produce substantially higher ECE values relative to the uncalibrated models.

In contrast, traditional architectures demonstrate more consistent responses to calibration techniques. ResNet-50 and ViT show calibration improvements across both benchmarks, though the magnitude of improvement is notably higher on ImageNet-V2 than ImageNet-A. This reflects the increasing challenge of calibration under severe distribution shifts. Nevertheless, the absolute ECE values remain lower for foundation models compared to traditional architectures, even under se-

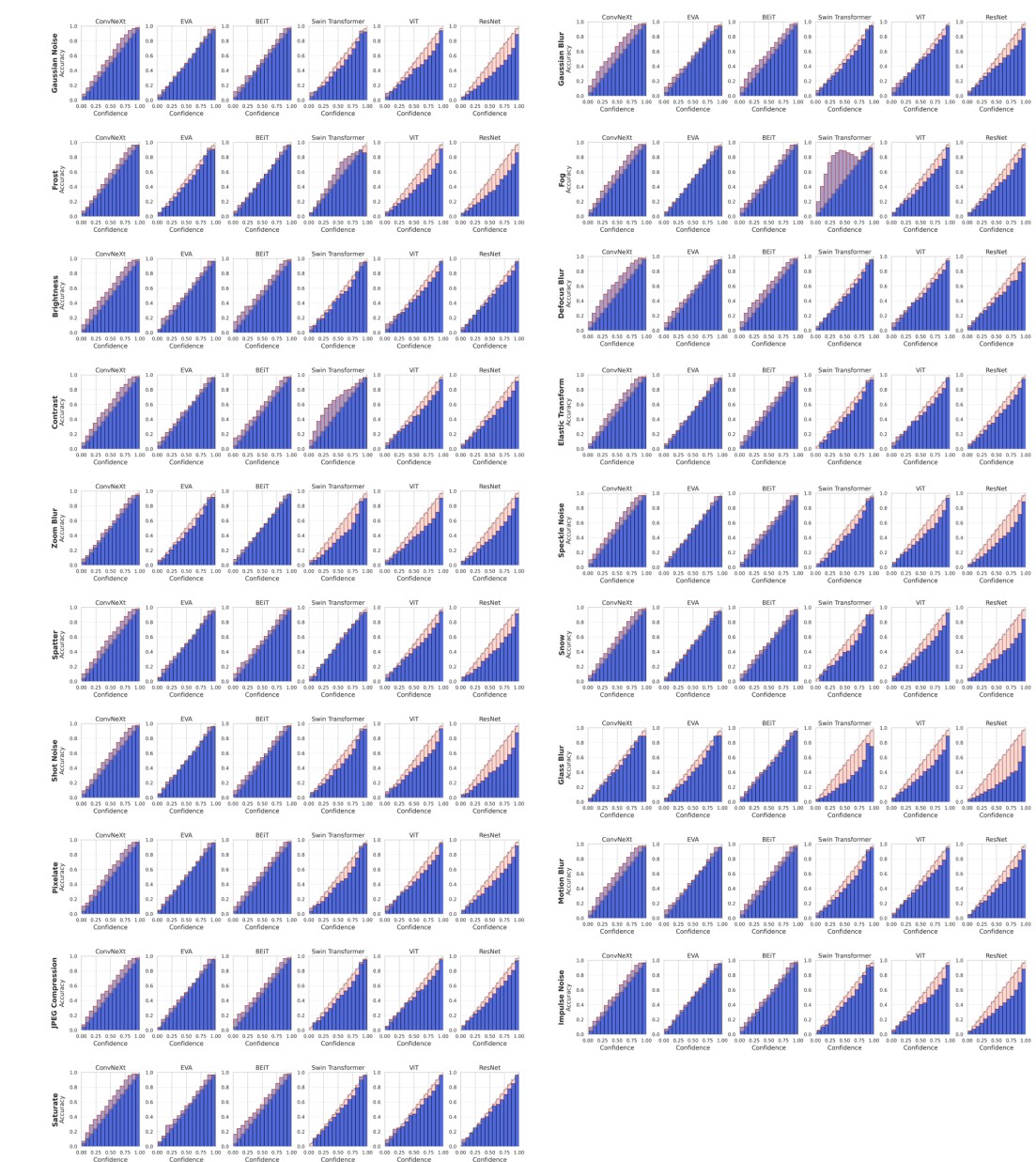

Figure 12: Individual reliability diagrams for all 19 synthetic corruptions of ImageNet-C for severity 3.

vere shifts. Interestingly, the Swin Transformer also exhibits negligible responsiveness to post-hoc calibration across real-world distribution shifts, reaffirming the pattern observed in in-distribution scenarios. This consistent behavior suggests architectural characteristics that fundamentally limit the effectiveness of post-hoc calibration techniques.

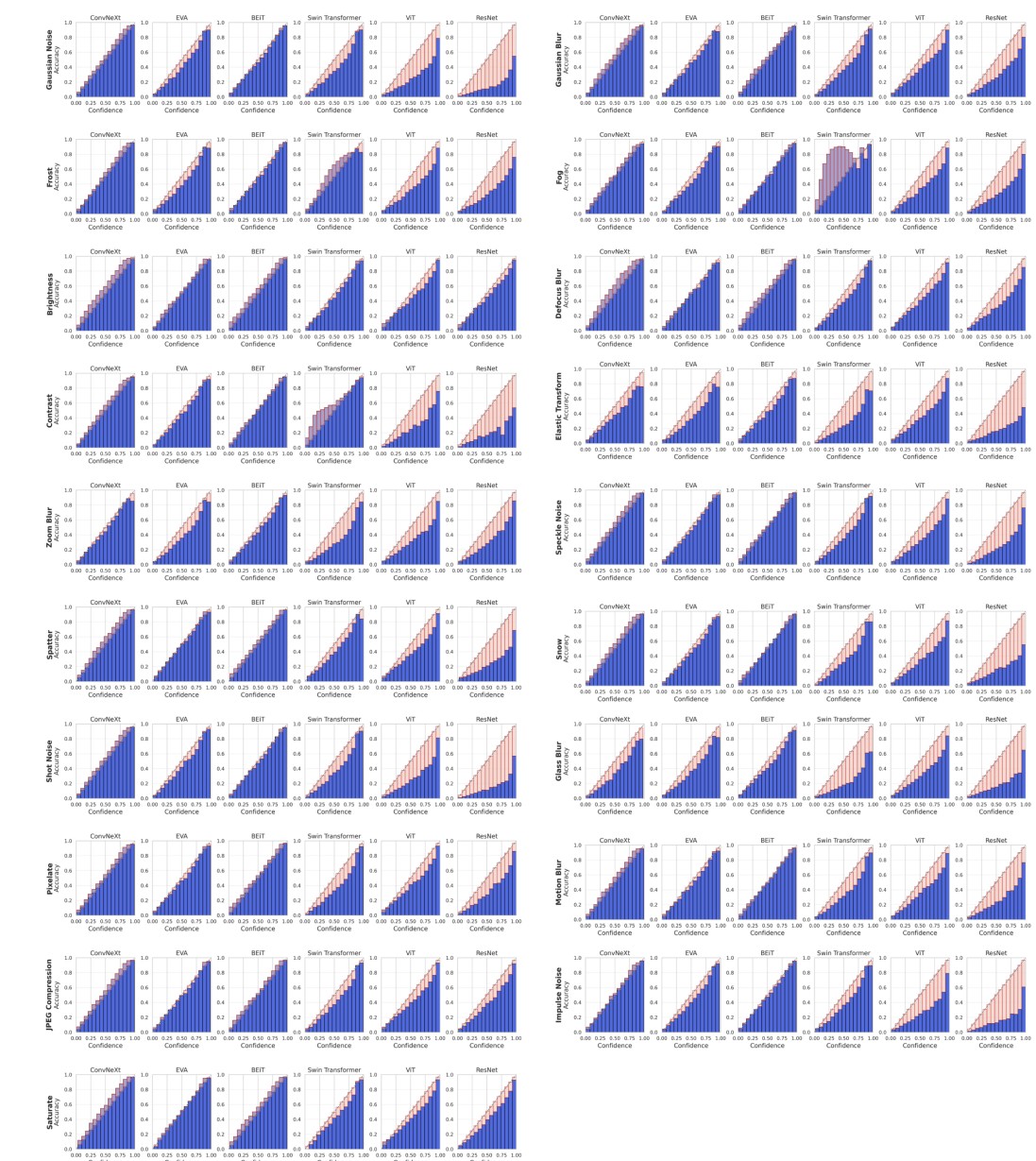

Figure 13: Individual reliability diagrams for all 19 synthetic corruptions of ImageNet-C for severity 5.

# H    LLM USAGE STATEMENT

Large Language Models (LLMs) were used exclusively for linguistic refinement and proofreading of this manuscript. Specifically, we employed LLMs to improve grammar, sentence structure, and overall readability of the text. No LLMs were used for research design, hypothesis generation, data analysis, interpretation of results, or the development of core ideas presented in this work. All scientific contributions, experimental designs, theoretical insights, and conclusions are entirely the product of the authors' original research.

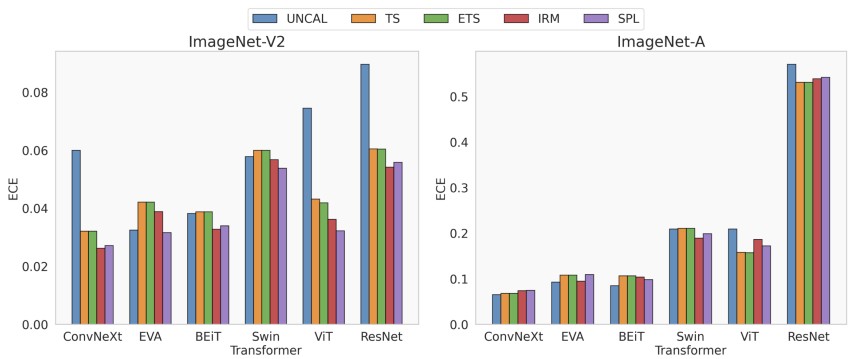

Figure 14: Analysis of post-hoc calibration methods under real-world distribution shift.

# I  CODE AVAILABILITY

We implement and analyze the post-hoc calibration methods introduced in the previous section within a newly developed Python package, called ModelTransformer. This package provides a unified framework inspired by the design principles of scikit-learn. The package offers consistent interfaces for fitting and transforming data, enabling parameter estimation on validation datasets and subsequent application to test sets. The complete implementation is available at `https://github.com/XXX/XXX/`.

All the code used to generate the analysis and figures in this paper is publicly available at `https://github.com/XXX/XXX/`. This repository contains the code that enables the complete reproduction of our experimental results and graphical representations.

# J  DATA AVAILABILITY

The complete set of raw and recalibrated model outputs used in this paper is publicly available at `https://doi.org/XX.XXXX/zenodo.XXXXXXXX`.

This extensive collection of datasets enables the full reproduction of our calibration analysis, as well as providing opportunities for researchers to conduct further investigations beyond the scope of this work.

