# OpenReview forum: "Beyond Overconfidence: Rethinking Calibration in Large-Scale Vision Models"
_ICLR.cc/2026/Conference — Submitted to ICLR 2026_

### Official Review · Reviewer_Sewz · 2025-10-28

**Soundness:** 3
**Presentation:** 3
**Contribution:** 2
**Rating:** 4
**Confidence:** 4

**Summary:**

The paper evaluates several state-of-the-art vision models on ImageNet-1k and its distribution-shifted variants (ImageNet-C, ImageNet-A, ImageNet-V2), assessing both calibration—measured via top-label Expected Calibration Error (ECE)—and accuracy. The six models considered are ResNet, ViT, Swin, BEiT, EVA, and ConvNeXt. The results show that **ConvNeXt, EVA, and BEiT tend to be under-confident** in-distribution but exhibit improved calibration under distribution shifts, whereas ResNet, ViT, and Swin display the opposite pattern, being better calibrated in-distribution but at the cost of lower accuracy.

The authors then investigate two factors that may influence this calibration behavior. First, for a fixed ViT architecture, they show that switching the pretraining objective from supervised to contrastive learning improves accuracy but leads to under-confidence. Second, for a fixed ViT or ResNet architecture, they find that applying advanced regularization and data augmentation techniques similarly improves accuracy while creating under-confidence.

Finally, the authors demonstrate that this miscalibration can be effectively mitigated by standard post-hoc calibration methods, although these techniques remain less effective under severe distribution shifts.

**Strengths:**

* The paper is clearly written and easy to follow.

* I have seen results showing that base LLMs tend to be under-confident (e.g., [Cruz2024]), while their instruction-tuned versions become over-confident. However, I am not aware of prior work reporting that models such as ConvNeXt, EVA, and BEiT are under-confident in-distribution and better calibrated under distribution shifts. To my knowledge, this observation is therefore novel.

[Cruz 2024] Cruz et al, Evaluating language models as risk scores

**Weaknesses:**

I think the significance and potential impact of the paper are limited by the relatively narrow scope of the experimental study, which evaluated the ECE of 6 models + 4 variants.

In line 462, the authors mention that the paper focuses on diagnostics (e.g., identifying under-confidence in certain models) rather than uncovering its causes. However, given ICLR’s standards, it would be reasonable to expect a broader experimental exploration in this direction, across architectures, model sizes, pretraining objectives, regularization strategies, fine-tuning, etc (or at least a subset of these.) I acknowledge that Section 4.3 provides preliminary insights, and they are very valuable. But again for ICLR standards, I would expect deeper insights in this direction.

**Questions:**

- Did the authors train any of the evaluated models, or are they all available on Hugginface? If yes, could the authors provide the links to the models used?
- In section 4.2, the authors state that large-scale models exhibit systematic in-distribution underconfidence, encompassing ConvNext, BEiT & EVA in the large scale models. However, why is the ViT not considered as large scale, since it is also pretrained on ImageNet-21k, and is (I would say) not smaller than the other models ?

---

> ### Author Response · Authors · 2025-11-21
> **Authors' Response**
>
> Thank you for the thoughtful review and for recognizing the novelty of our observations about underconfidence in modern large-scale vision models. We address your concerns below.
>
> ---
>
> **W1 Limited Scope and Broader Experimental Exploration**
>
> Thank you for this insightful comment. In direct response, we **substantially broadened our experimental analysis** and now provide a much **deeper investigation into the underlying causes of underconfidence**.
> We conducted a **comprehensive study on the ViT architecture**, which is an especially meaningful setting given that prior work consistently describes ViTs as overconfident. Our new experiments span **1,008 independently trained ViT models**, varying key training pipeline components along the lines of your suggestion: pretraining corpus, augmentation strength (RandAugment, Mixup), stochastic depth, dropout, weight decay, and fine-tuning learning rate.
>
> The results reveal a clear and robust mechanism:
> * **Stronger augmentations consistently induce underconfidence** across all ViT sizes (B/16, B/32, L/16, S/16, S/32, Ti/16).
> * **Pretraining amplifies this effect**, shifting models from overconfident (no pretraining) to more underconfident with pretraining.
>
> These results confirm that training strategy choices (not architectural design) are the dominant drivers of modern underconfidence.
> A visual summary of these findings is now included in Fig. 2 c and d.
> We believe this significantly strengthens the paper’s contribution and aligns well with ICLR’s expectations for depth and rigor.
>
> ---
>
> **Q1 Employed Models**
>
> Thank you for the question. All evaluated models were publicly available; none of them were trained by us. Our goal was to rely on high-quality, thoroughly validated checkpoints that use the full set of training techniques (e.g., large-scale pretraining, advanced regularization, strong augmentation). Training these models ourselves might not achieve comparable accuracy and could introduce confounds due to differences in training pipeline quality.
> As noted in Appendix F and Table 1 of the submitted paper, we already list the exact model specifiers used for all architectures. These uniquely identify the checkpoints employed in our experiments. To improve clarity, we have now **added an explicit reference to Appendix F in Section 4.1** of the revised manuscript and **included direct HuggingFace URLs** for immediate access to the model cards.
>
> ---
>
> **Q2 Definition of Large-Scale Models**
>
> Thank you for the thoughtful question. In our paper (Section 1), we define large-scale models as those that combine massive pretraining with modern large-scale training techniques. This is an aspect that recent architectures such as ConvNeXt, EVA, and BEiT heavily rely on. While the original ViT (Dosovitskiy et al., 2021) was indeed pretrained on ImageNet-21k and is comparable in size, it did not employ many of the training strategies that have since become standard for large-scale models, such as strong augmentation pipelines (e.g., RandAugment, Mixup, CutMix), advanced regularization, or optimized training recipes.
>
> Importantly, the distinction is not architectural. As we show in Section 4.3, when the same ViT architecture is trained with a modern augmentation–regularization pipeline (ViT-AugReg), it exhibits the same systematic underconfidence and calibration characteristics as the other large-scale models in our study (ConvNeXt, EVA, BEiT). This reflects our main conclusion: that calibration behavior is driven primarily by training methodology rather than by architectural scale or type alone.

---

### Official Review · Reviewer_GXKG · 2025-10-31

**Soundness:** 2
**Presentation:** 2
**Contribution:** 3
**Rating:** 4
**Confidence:** 3

**Summary:**

This paper systematically evaluates the uncertainty calibration performance of three large-scale vision models—ConvNeXt, EVA, and BEiT—and reveals findings that contradict prior research conclusions. The study shows that these models generally exhibit systematic underconfidence on in-distribution data, instead of the commonly reported overconfidence of tranditional models, leading to higher calibration errors. What's more, their calibration error decreases under out-of-distribution conditions. Furthermore, the authors demonstrate that these models respond well to post-hoc calibration methods (e.g., Temperature Scaling) in in-distribution scenarios. Nonetheless, the effectiveness of such methods degrades significantly under severe distribution shifts.

**Strengths:**

1. The overall structure of the paper is clear and well-organized.

2. The related work section clearly articulates how this paper differs from previous studies, allowing readers to quickly grasp the central contributions.

3. The experiments are diverse and conducted across multiple architectures and datasets. Figures 4(a) and 6 effectively correspond to and support the main conclusions.

**Weaknesses:**

1. There are several typos and grammatical errors throughout the paper. The authors should carefully proofread the manuscript to meet publication standards. For example, there is a mistake "Eq. ??" in line 135.

2. Some of the paper’s conclusions are not clearly stated. For instance, Section 4.4 only provides descriptive commentary on figures without drawing any conclusions. If the intended point is that “Figure 3 shows architecture-specific differences in effectiveness,” this claim is neither visually evident from the figure nor supported by textual explanation. Similarly, while Figure 4 is later discussed, the meaning of the solid and dashed markers and the significance of their connecting lines are not made clear from the figure itself, although in the latter context.

3. Certain conclusions lack sufficient evidence or persuasiveness. For example, the findings in Section 4.3 are only validated on a single model, which limits their generality. In Section 5, the final key insight labeled as “best” actually refers only to being the best among the three evaluated models, rather than a universal best.

**Questions:**

Why did the authors choose to evaluate these three models rather than using larger-scale models such as LLaVA or Qwen-VL?

---

> ### Author Response · Authors · 2025-11-21
> **Authors' Response**
>
> We thank the reviewer for the constructive feedback and for acknowledging the clarity and rigor of our work. We appreciate the detailed suggestions, which have helped us improve the manuscript. Below, we address each point raised.
>
> ---
>
> **W1: Typos**
>
> Thank you for pointing this out. We have thoroughly proofread the manuscript and corrected all identified typographical and grammatical issues. The incorrect reference ("Eq. ??" on line 135) has also been fixed.
>
> ---
>
> **W2: Clarification of Figures 3 and 4**
>
> Thank you for the suggestion. We have revised Section 4.4 and the caption of Figure 3 to clarify the conclusion.
> Furthermore, we have updated the caption of Figure 4 to clarify the meaning of the markers and lines.
> Specifically, green markers indicate large-scale models, red markers indicate traditional models, and gray markers show in-distribution performance on the clean ImageNet test set. Dotted lines connect each model’s in-distribution result to its corresponding out-of-distribution result, allowing visualization of the effect of distribution shift.
>
> ---
>
> **W3: Lack of Sufficient Evidence**
>
> Thank you for raising this important point. We agree that the original version of Section 4.3 did not provide sufficiently broad evidence to support our conclusions. In direct response to your feedback, we substantially expanded the analysis and now include a large-scale ablation study designed to rigorously test the generality of the claimed effect.
>
> Specifically, we conducted a comprehensive investigation on the ViT architecture, which is an especially relevant setting given that prior work consistently describes ViTs as overconfident. Our new study spans **1,008 independently trained ViT models**, systematically varying key components of the training pipeline (pretraining corpus, augmentation strength, Mixup, stochastic depth, dropout, weight decay, and fine-tuning learning rate).
>
> The expanded results reveal a consistent and robust pattern:
>
> * **Stronger augmentations consistently increase underconfidence** across all ViT variants we evaluated (B/16, B/32, L/16, S/16, S/32, Ti/16).
> * **Pretraining amplifies this effect:** models without pretraining tend to be overconfident, whereas pretrained models become increasingly underconfident.
>
> We now include a consolidated visualization of these findings in Fig. 2 c and d. We hope that this substantially strengthened evidence addresses your concern and clarifies the generality of our conclusions.
>
> **Regarding Section 5:**
> We would like to clarify that our conclusion is intended strictly within the scope of the models we evaluated. The submitted version already states this explicitly ("among the evaluated models"), but we agree that the strength of the supporting evidence may not have been sufficiently emphasized.
>
> To avoid any misunderstanding, we highlight that this conclusion is based on a **broad and consistent evaluation across six models**, covering (i) the clean in-distribution ImageNet test set, (ii) 95 distribution-shifted datasets from ImageNet-C, and (iii) two real-world distribution shifts (ImageNet-V2 and ImageNet-A)
>
> ConvNeXt ranked best within our evaluation setup across all these settings - accuracy, calibration, and post-recalibration quality, leading to our phrasing of it as the strongest option among the evaluated models.
>
> ---
>
> **Q1: Model Selection**
>
> Thank you for the question. Our study focuses on **pure vision classification**, where models produce probabilistic class outputs suitable for **conventional calibration metrics** (see Section 3, Problem Definition).
>
> The 6 models investigated in detail cover (i) representative top-performing vision architectures from recent model families (EVA, BEiT and ConvNeXt), and (ii) representative earlier architectures (Swin Transformer, ViT, ResNet) to allow comparisons across architectural generations and training regimes
>
> Models such as **LLaVA** and **Qwen-VL** are **multimodal vision–language models**. They do not produce class logits over a fixed label set but generate free-form text using a large language model. Therefore, they **cannot be evaluated or recalibrated using conventional calibration techniques**, making them incompatible with the problem formulation of this work.

---

### Official Review · Reviewer_Hc1C · 2025-11-01

**Soundness:** 2
**Presentation:** 2
**Contribution:** 2
**Rating:** 2
**Confidence:** 3

**Summary:**

The paper investigates how miscalibration differs under various distribution shifts, showing that models tend to be *underconfident* for in-distribution data but become *overconfident* under out-of-distribution (OOD) shifts.

**Strengths:**

**Pros**

* It is valuable to examine calibration performance across different data groups, such as in-distribution and out-of-distribution scenarios.
* The writing is clear to follow.

**Weaknesses:**

**Cons**
* Missing important related works with similar findings [1] or relevant methods for reducing miscalibration across different groups [1][2].
* Lacks theoretical analysis or insights explaining the observed calibration behaviors.
* The evaluated models are still limited in scope compared to prior comprehensive studies [1][3].
* The overall contribution is limited. Most findings have already been identified in prior works, such as miscalibration under group shifts [1] and the accuracy–calibration trade-off [3].


---

**References**

[1] Xiong, Miao, et al. *"Proximity-informed calibration for deep neural networks."* NeurIPS 2023

[2] Perez-Lebel, Alexandre, Marine Le Morvan, and Gaël Varoquaux. *"Beyond calibration: estimating the grouping loss of modern neural networks."* ICLR 2023

[3] Minderer, Matthias, et al. *"Revisiting the calibration of modern neural networks."* NeurIPS 2021

**Questions:**

Please see **Weaknesses**.

---

> ### Author Response · Authors · 2025-11-21
> **Authors' Response (Part1)**
>
> We thank the reviewer for the feedback and for acknowledging the importance of our study and the clarity of our writing.
>
> We would like to address the main concerns raised and clarify several points where we believe there may be misunderstandings about our contributions.
>
> ---
>
> **C1: Missing related work**
>
> We thank the reviewer for highlighting the missing discussion of group-wise calibration methods and uncertainty decomposition. In the revised manuscript, we expanded the related work section accordingly.
>
> We now provide a detailed discussion of proximity-based calibration [1] and related density-aware approaches [a] in Section 2 (Post-hoc Calibration Techniques). In addition, we added a new paragraph on uncertainty decomposition in Section 2, incorporating [2] and other foundational work in this area.
>
> Finally, regarding the comment on "similar findings," we clarify the relationship between our results and [1] in our response to Comment C4, highlighting the distinct contributions of our work.
>
>
> ---
>
> **C2 + C3: Lack of Theoretical Analysis/Insights And Limited Scope**
>
> We appreciate this constructive feedback. We agree that the original submission offered limited insight into the mechanisms behind our observations. To address this, we substantially expanded our analysis with a large-scale ablation study designed specifically to deepen our understanding of the observed phenomenon.
>
> We conducted a comprehensive investigation of Vision Transformers (ViTs), a particularly relevant architecture given that prior work consistently describes ViTs as overconfident. Our expanded study comprises **1,008 independently trained ViT models**, systematically ablating seven key training factors: pretraining corpus, augmentation and Mixup strength, stochastic depth, dropout, weight decay, and fine-tuning learning rate. The evaluation spans six variants (B/16, B/32, L/16, S/16, S/32, Ti/16).
>
> Across all configurations, we observe a consistent and robust pattern:
>
> - **Augmentation-induced underconfidence:** Stronger augmentation strategies consistently shift models toward underconfidence
> - **Pretraining amplifies the effect:** Models trained from scratch exhibit overconfidence, while pretrained models demonstrate progressively stronger underconfidence.
>
> These findings are now presented in consolidated form in Figure 2 c and d of the revised manuscript.
>
> *Theoretical Perspective:*
>
> Beyond the expanded empirical study, we also provide a theoretical perspective grounded in Vicinal Risk Minimization (VRM) that helps explain why stronger augmentations (and pretraining) systematically induce underconfidence. VRM (Chapelle et al., Vicinal Risk Minimization, NeurIPS 2001) minimises the expected loss not only on the empirical samples 𝑥 but also on a neighbourhood 𝑣(𝑥) induced by data augmentation: $$L_{\text{VRM}} = E_{(x,y)∼𝒟} E_{x'∼v(x)} ℓ(f(x'), y) \ .$$
>
> Therefore, the model is trained not on the empirical distribution but on a vicinal distribution that assigns probability mass to neighborhoods around each training sample. When augmentations become stronger or more diverse (e.g., RandAugment, Mixup), the vicinal distribution becomes increasingly smoother and more dispersed, effectively enlarging the support region around each labeled example.
>
> This has two implications relevant to underconfidence:
>
> 1. **Softening of decision boundaries.** A wider vicinal distribution forces the classifier to assign similar probabilities to a broader set of augmented variants, which encourages smoother logits. This naturally pushes predictions toward underconfidence rather than overconfidence.
>
> 2. **Pretraining further amplifies VRM's smoothing effect.** Pretrained models already encode broader invariances and feature smoothness across large datasets. When combined with strong augmentations during fine-tuning, the model effectively samples from an even more dispersed vicinal distribution. This compounds the smoothing effect on logits, leading to stronger underconfidence, exactly what we observe empirically.
>
> This theoretical viewpoint helps explain the universal trends across all 1,008 ViT models we evaluated and supports the conclusion that underconfidence emerges from properties of the training strategy, not from architectural biases.
>
> [a] Tomani et al., Beyond in-domain scenarios: Robust density-aware calibration, ICML 2023.

---

> ### Author Response · Authors · 2025-11-21
> **Authors' Response (Part 2)**
>
> **C4: Limited Contribution**
>
> We respectfully disagree with the assessment that our contributions are limited or largely overlapping with prior work. We believe this conclusion might arise from a misunderstanding of the core differences between our findings and those in [1] and [3]. Below, we provide a direct comparison.
>
>
> **Summary of Key Contributions and Coverage in Prior Work**
>
> | Our Key Contribution | Addressed in Prior Work? |
> |-------------|--------------------------|
> | Systematic underconfidence in modern large-scale models | No - prior work overwhelmingly reports overconfidence |
> | Attribution of calibration behavior to training methodology, not architecture | No - [3] attributes calibration primarily to architecture |
>
> **Key Distinctions from [1] (Proximity Bias)**
>
> The reviewer suggests that [1] reports "similar findings." We respectfully clarify that the phenomena are fundamentally different:
>
> **What [1] studies**
> - Models exhibit different calibration errors for high-proximity vs. low-proximity samples **within the same confidence bin**
> - Low-proximity samples are more overconfident than high-proximity samples
> - Focus: **relative miscalibration** between subgroups
>
> **What we study**
> - Modern large-scale models exhibit systematic underconfidence **across all samples**
> - This contradicts decades of overconfidence findings (Guo et al. 2017; prior work)
> - Focus: **absolute change** in calibration behavior
>
> Thus, [1] documents a local subgroup effect, while we identify a global shift in calibration properties driven by training methodology.
>
> **Key Distinctions from [3]**
>
> The reviewer suggests our findings are similar to [3]. However, several of our core results directly contradict the conclusions of [3]:
>
> **1. Calibration direction**
> - **[3]:** "Most models are slightly overconfident in-distribution."
> - **Ours:** Modern large-scale models are systematically underconfident in-distribution (Fig. 1b, Sec. 4.2).
> - → This shift has not been documented previously.
>
> **2. Calibration quality**
> - **[3]:** "Modern image models are well calibrated" and "there is no general trend for recent or highly accurate neural networks to be poorly calibrated."
> - **Ours:** Accuracy-calibration trade-off emerges with large-scale pretraining and aggressive regularization/augmentation (Fig. 1a).
> - → Calibration worsens in the regime where accuracy improves.
>
> **3. Role of architecture**
> - **[3]:** "Architecture is a major determinant of calibration properties."
> - **Ours:** Training methodology is the dominant factor. Controlled experiments show that the same architecture (e.g., ResNet-50, ViT) can move from overconfident to strongly underconfident depending solely on the training pipeline (Sec. 4.3, Fig. 2).
> - → Calibration is not an architectural property but a consequence of data scale, pretraining, and augmentation strength.

---

### Author Response · Authors · 2025-11-21
**General Response to Reviewer Feedback**

We thank the reviewers for their constructive feedback and for **highlighting the importance** (Hc1C), **novelty** (Sewz), and **clarity** (Hc1C, GXKG, Sewz) of our study. In our response, we have aimed at addressing in full the reviewers' concerns. Based on their suggestions, we have substantially strengthened the paper along three key dimensions:

---

**More models and empirical evidence:**

To further strengthen our analysis on the causes underlying the observed underconfidence, we added a large-scale ablation study with 1,008 ViT models of different sizes as well as different pretraining, regularization, augmentation and fine-tuning strategies. This further supported our conclusions and revealed a consistent mechanism behind underconfidence: stronger augmentations induce underconfidence, and pretraining consistently amplifies this effect across all ViT variants. With this analysis we have widened the scope of the original analysis as suggested by the reviewers and established that the effect generalizes across a wide range of models and training setups.

**A theoretical clarification:**

We provide a concise VRM-based explanation showing how stronger augmentation and pretraining lead to smoother vicinal distributions and systematically lower confidence, thereby supporting our empirical findings.

**Improved positioning and clarification of contribution:**

We expanded the related work and clarified the key distinctions from prior work. Our central contributions - identifying and characterizing a shift toward underconfidence of modern models, pinpointing training methodology as the primary cause (rather than architecture as previously suggested), and analyzing post-hoc calibration behaviour under distribution shift - are not addressed in earlier literature.

---

We believe these revisions substantially strengthen the rigor and contribution of the paper and address all major concerns.

---

### Author Response · Authors · 2025-12-03
**Concluding Remarks**

In our study, we investigate the uncertainty calibration behavior of modern large-scale vision models and the effectiveness of post-hoc calibration methods.

We present three key findings that challenge the current understanding of model calibration: (a) contemporary large-scale models exhibit systematic underconfidence, reversing the overconfidence paradigm that has defined neural network calibration research since Guo et al. (2017);  (b) these underconfident models exhibit robust calibration properties under distribution shift (contrary to Ovadia et al., NeurIPS 2019); (c) training methodology rather than architecture determines calibration behavior (contrary to Minderer et al., NeurIPS 2021). These findings represent a fundamental shift in calibration behavior with direct implications for deploying large-scale models in practice.

In their constructive reviews, reviewers have raised two major common points that we addressed as summarized below. No reviewer had replied before comments were closed on November 25th.

- **Lacks Theoretical Analysis (Reviewer Hc1C, Sewz):** We added a **theoretical analysis based on Vicinal Risk Minimization (VRM)** in Section 4.3, explaining why aggressive augmentation and large-scale pretraining produce smoother, less confident predictions.

- **Limited Model Scope (Reviewer Hc1C, Sewz, GXKG):** We conducted an **ablation study with 1,008 independently trained ViT models**, substantially exceeding the coverage of prior work (e.g., Minderer et al. (2021) and Xiong et al. (2023)). For these models, pretraining, augmentation, regularization, and fine-tuning were systematically varied across six architectural variants. This further supported our conclusion on the influence of training methodology on uncertainty calibration. The results also empirically validate the VRM analysis.

In addition, we also addressed the novelty concern which was raised by Reviewer Hc1C as a major concern:

- **Limited Novelty/Contribution (Reviewer Hc1C):** We respectfully disagree with the assessment that our findings overlap with prior work. Our results **contradict rather than confirm established calibration research**. A detailed comparison with Minderer et al. (2021) and Xiong et al. (2023) is provided in our rebuttal.

Furthermore, we addressed all remaining minor concerns as summarized below:

| Minor Concern | Reviewer | How Addressed |
|---------------|----------|---------------|
| Missing related work | Hc1C | Added Xiong et al. (NeurIPS 2023) and Perez-Lebel et al. (ICLR 2023) to Section 2 (Lines 106–110 and 122–125) |
| Unclear conclusions and captions | GXKG | Revised Section 4.4 (Lines 357–366); improved figure captions for Figures 3 and 4 |
| Typos and broken references | GXKG | Thoroughly proofread and corrected |
| Why not LLaVA/Qwen-VL? | GXKG | Clarified scope additionally in Section 5 (Lines 466–469) |
| Why is ViT not "large-scale"? | Sewz | Clarified the definition in Section 4.1 (Lines 185–190): the distinction concerns the training methodology, not the architecture |
___
We would like to once again thank the reviewers for their thoughtful and constructive feedback, which has helped us further strengthen the paper. We believe that the revised manuscript now addresses all concerns.

---

### Meta-Review · Area_Chair_KFD1 · 2026-01-06

**Summary:**

The reviewers raised several key concerns: the lack of theoretical analysis to explain the observed calibration behaviors; the limited scope of models evaluated compared to prior comprehensive studies, which could affect the generalizability of the findings; questions regarding the novelty and contribution of the work, with some reviewers suggesting significant overlap with existing literature; issues with clarity in figures, conclusions, and manuscript presentation, including typos and unclear captions; and queries about the definition of "large-scale" models and the rationale behind the specific model selection for the study.

**Reviewer Concerns:**

The authors' rebuttal effectively addressed several major concerns: they introduced a theoretical explanation based on Vicinal Risk Minimization (VRM) to clarify the mechanism behind underconfidence; they expanded the experimental scope with a large-scale ablation study. However, the concern regarding the fundamental novelty of the contribution, specifically, whether the identified shift from overconfidence to underconfidence constitutes a sufficiently distinct or transformative finding, remains an outstanding concern.

**Reviewer Scores:**

Reviewer HctC would likely keep the score (Reject): acknowledging the strengthened theoretical and empirical foundations but potentially remaining unconvinced about the overarching novelty.

Reviewer GXKG would likely increase the score from 4 to 6, as the concerns were addressed.

Reviewer Sewz would likely increase the score from 4 to 6, as the authors' large-scale ablation study responded to their primary concern about experimental scope and provided deeper insight into the causes of underconfidence.

---

### Decision · Program_Chairs · 2026-01-26

Reject